# A Humanized CB1R Yeast Biosensor Enables Facile Screening of Cannabinoid Compounds

**DOI:** 10.3390/ijms25116060

**Published:** 2024-05-31

**Authors:** Colleen J. Mulvihill, Joshua D. Lutgens, Jimmy D. Gollihar, Petra Bachanová, Caitlin Tramont, Edward M. Marcotte, Andrew D. Ellington, Elizabeth C. Gardner

**Affiliations:** 1Center for Systems and Synthetic Biology, Department of Molecular Biosciences, The University of Texas at Austin, Austin, TX 78712, USActramont@utexas.edu (C.T.);; 2Antibody Discovery and Accelerated Protein Therapeutics, Center for Infectious Diseases, Houston Methodist Research Institute, Houston, TX 77030, USA; jgollihar2@houstonmethodist.org; 3Department of Bioengineering, Rice University, 6100 Main St., Houston, TX 77005, USA

**Keywords:** cannabinoid receptors, humanized yeast, G-protein-coupled receptors

## Abstract

Yeast expression of human G-protein-coupled receptors (GPCRs) can be used as a biosensor platform for the detection of pharmaceuticals. Cannabinoid receptor type 1 (CB1R) is of particular interest, given the cornucopia of natural and synthetic cannabinoids being explored as therapeutics. We show for the first time that engineering the N-terminus of CB1R allows for efficient signal transduction in yeast, and that engineering the sterol composition of the yeast membrane modulates its performance. Using an engineered cannabinoid biosensor, we demonstrate that large libraries of synthetic cannabinoids and terpenes can be quickly screened to elucidate known and novel structure–activity relationships. The biosensor strains offer a ready platform for evaluating the activity of new synthetic cannabinoids, monitoring drugs of abuse, and developing therapeutic molecules.

## 1. Introduction

GPCRs are common therapeutic targets, and an estimated 30% of FDA-approved drugs target this family of receptors [1]. The native pheromone response pathway of Saccharomyces cerevisiae is regulated by an endogenous GPCR (Ste2/Ste3) [2], and the yeast receptors can be replaced with human GPCRs, leading to signal transduction via modified Gpa1 Gα proteins and the activation of the mitogen-activated protein kinase (MAPK) cascade, ultimately inducing gene expression via the transcription factor Ste12 [3]. A family of GPCRs with medical and industrial importance is cannabinoid receptor type 1 (CB1R). The most abundant GPCR in the brain [4], CB1R is activated by the psychoactive drug tetrahydrocannabinol (THC) and is also the target of the endocannabinoids 2-arachadonoylglycerol (2-AG) and anandamide (AEA) [5]. These neurotransmitters exist as lipid precursors embedded in cell membranes where they are cleaved by lipases and liberated for receptor activation [5]. Endocannabinoid regulation via CB1R is implicated in neuronal excitability, where retrograde transmission of endocannabinoids from postsynaptic cells activates CB1R on presynaptic neurons and negatively regulates presynaptic neurotransmission via the Gαi/o pathway [6]. CB1R dysregulation is, in turn, associated with schizophrenia [7].

Synthetic cannabinoids have been developed to elicit responses by cannabinoid receptors. For decades, these drugs have been sold illicitly to consumers looking for similar psychoactive effects as THC. Synthetic cannabinoids are quite popular, being the second-most used illegal substance by young adults [8]. Unlike THC, however, these compounds are typically not identified in conventional drug screens. Interestingly, many synthetic cannabinoids are much tighter binders and subsequently agonists to cannabinoid receptors than THC and are often sold as mixtures uncharacterized for human use [8]. Together, the high potency and lack of regulation of these compounds have led to many cases of adverse effects from recreational use, including acute psychosis, seizures, dependence, and death [8,9,10,11]. Governments have attempted to regulate these compounds, but regulation remains a challenge as new compounds and analogs of existing ones are created frequently that evade restriction [8].

Given the role of cannabinoids in the treatment of chronic pain, epilepsy, and psychiatric disorders, there is a growing demand for next generation cannabinoid medicines. Ideal therapeutic candidates should activate Cannabinoid receptor type 2 (CB2R) while avoiding potent activation of CB1R and triggering subsequent psychoactive effects. Discrimination between the two receptors is challenging, as CB1R and CB2R share a high degree of sequence similarity, including highly conserved binding pockets [12].

Cannabinoid biosensor yeast strains have the potential to serve as a rapid, inexpensive, and robust screening platform. While such CB2R yeast strains have been recently developed [13,14], no CB1R strain has been previously described. Herein, we demonstrate the engineering of CB1R yeast biosensors by combining synthetic biology approaches that target receptor trafficking and membrane composition. Using these optimized strains, we screened more than 400 synthetic cannabinoids and terpenes, characterizing known effectors and discovering unknown functions of cannabinoids, including analogs of controlled drugs of abuse. Our CB1R biosensor strain provides a rapid functional screen that can be used to readily rationalize structure–activity relationships at this receptor and should accelerate the development of safe cannabinoid therapeutics into the future.

## 2. Results

### 2.1. Engineering Cannabinoid Receptor Function in Yeast

While several human GPCRs have previously been functionally expressed in yeast [15,16], CB1R has remained recalcitrant to functional expression in the orthogonal host. To adapt yeast for CB1R compatibility, we generated a modified strain which enables human GPCRs to couple with the signal transduction machinery of the pheromone response pathway (Figure 1a). The yeast of the MATa haplotype uses the Ste2 GPCR to detect pheromones and drives the expression of genes related to mating through the MAP kinase pathway [2]. Previous work has shown that Ste2 may be functionally replaced with human GPCRs [17]. Additionally, the yeast G-alpha protein Gpa1 can efficiently transduce signals from human GPCRs when its C-terminus, which directly contacts the receptor, is modified to match the corresponding human G-alpha protein [3]. Since CB1R typically interacts with human Gαi/o proteins [18], we used a Gpa1-Gɑi3 chimera with a humanized C-terminus (-ECGLY). The knockout of the GTPase-activating protein Sst2 [19] and cell-cycle arrest regulator Far1 further augment biosensor utility [3] by enhancing pathway sensitivity and allowing continued growth of stimulated yeast, respectively. Finally, replacing the pheromone response gene Fig1 with the fluorescent reporter ZsGreen [20] enables facile screening of activated GPCRs as the MAP kinase pathway drives fluorescent protein expression from the pFig1 promoter.

Initially, we used our engineered ZsGreen reporter strains to test the expression of the wild type CB1R receptor via fluorescence. In brief, we incubated strains with a cannabinoid or vehicle control (DMSO, 5% *v*/*v*) for 8 h and measured the ZsGreen via fluorescent cytometry, where we examined the mean fluorescence of a population of cells in the presence of ligands. The wild-type, leaderless CB1R showed low (<4-fold) signaling with the synthetic cannabinoid Arachidonyl-2′-chloroethylamide (ACEA) (Figure 1b), which we speculated might have been due to impaired plasma membrane localization. To improve basal function, we modified the popular MoClo yeast toolkit parts library [21] to include yeast-optimized pre-pro signals (designated type 3a) after the type 2 promoter and before the type 3 GPCR, yielding seamless fusions. Human CB1R also has a long N-terminal domain that can block co-translational insertions into the endoplasmic reticulum [22], thus partially redirecting the receptor to localize in the mitochondria [23]. To better specify transport, we created an N-terminally truncated version of the CB1 receptor, both with and without the syn-pre-pro secretion sequence. The syn-pre-pro modification alone improved activation approximately 7-fold, while the truncated receptor showed 15-fold activation; both modifications together yielded 28-fold activation (Figure 1b). However, the syn-pre-pro-Δ89-CB1R was found to have an EC50 of 147 nM, a 3.8-fold drop in sensitivity compared to the full-length, syn-pre-pro construct. Despite the tradeoff in sensitivity, the syn-prepro-Δ89-CB1R had a higher fraction of signaling cells in the population with a greater dynamic range and was subsequently chosen as the baseline CB1R biosensor strain (Figure 1d and Figure A1).

We then benchmarked this strain with the endocannabinoids 2-AG and AEA (Figure 1c), each with sensitivities similar to previously published values (AEA EC50, 4 nM; 2-AG EC50, 100 nM) [24]. To rule out the possibility of nonspecific cannabinoid signaling, we also compared ZsGreen fluorescence after 1 uM AEA in each of the four strains containing the syn-pre-pro-Δ89-CB1R, WT-CB1R, a non-cannabinoid serotonin receptor (5HT1AR), or a no-receptor strain (Figure 1d–e). Significant levels of signaling were observed in the CB1R strains, but not in the serotonin receptor or no-receptor controls. The functional improvement achieved by the combination of N-terminal truncation and the addition of a synthetic pre-pro leader highlighted that proper receptor trafficking can be an issue when heterologously expressing GPCRs in yeast. Since a high dynamic range was likely to be of the greatest utility for screens, especially with compounds with otherwise unknown affinities, we chose the combination of syn-pre-pro-Δ89-CB1R with ACEA to serve as the benchmark for further studies.

### 2.2. Yeast Sterol Composition Modulates Cannabinoid Receptor Activity

Cholesterol is necessary for the function of many GPCRs [25] and is co-crystallized in ~40% of GPCR PDB structures [26]. Cholesterol and its derivatives are hypothesized to negatively regulate CB1R via the cholesterol recognition amino acid consensus (CRAC) motif or another allosteric site [12,27,28,29]. The principal sterol of yeast, ergosterol, is structurally similar to cholesterol but with an additional methyl group and a pair of double bonds (Figure 2a). Despite significant structural overlap, our group has previously shown that the replacement of ergosterol with cholesterol can dramatically affect the sensitivity of human GPCRs in yeast. We therefore sought to clarify the role of ergosterol on cannabinoid receptor function.

Ergosterol and cholesterol share a common zymosterol precursor that can be converted to cholesterol through a four-enzyme pathway that produces seven downstream metabolites (Figure 2b). We recently replaced the native yeast sterol pathway with cholesterol biosynthesis enzymes to create a small library of yeast strains that produce varying levels of these seven metabolites in the GpaI-Gai3 chimeric ZsGreen human GPCR reporter strain [30]. The syn-pre-pro-Δ89-CB1R expression construct was transformed into fifteen strains (ST01–ST15, Figure 2c, Appendix A) selected to cover the metabolic and structural sterol space. We then performed dose response experiments with ACEA and measured the ZsGreen via fluorescent cytometry. We observed that all cholesterol intermediates had a broadly deleterious effect on CB1R signaling compared to ergosterol only (Figure 2c and Figure A2). The WT ergosterol strain had the highest fold change under fluorescence (93.6 ± 15.6 S.D.) versus the cholesterol/intermediate strains, which ranged from 34.3-fold (ST13) to 73.2-fold (ST01) (Figure 2c). The ergosterol strain also had the highest ACEA potency with an EC50 = 51 nM. The cholesterol intermediate strain ST07 had the next highest potency (EC50 = 65 nM) and the average EC50 value across the intermediate strains was 199 nM (Figure A2). These findings suggest that ergosterol does not negatively regulate the CB1R receptor as strongly as the native sterol cholesterol and its precursors. While the cholesterol producing strains may provide a more native environment for human CB1R, we reasoned that the increased sensitivity of the ergosterol strain may be advantageous for downstream applications in high-sensitivity biosensing.

### 2.3. High-Throughput Drug Screening of Terpenes with Dual Cannabinoid Receptor Biosensor Strains

The availability of a biosensor strain with an optimized cannabinoid receptor function immediately presents an opportunity to determine the relative activities of a variety of cannabinoids and other compounds. A compound library of over 300 synthetic cannabinoids was screened against the strain. A vehicle (DMSO, 1% *v*/*v*) or 1 μM inverse agonist rimonabant was included on each plate analyzed by flow cytometry. The compound library included known characterized agonists and structurally similar compounds; for example, AB-PINACA, ADB-PINACA, and AB-FUBINACA are all structurally related Schedule I controlled substances found in synthetic cannabis products. All test compounds were delivered at 1 µM, orders of magnitude in excess of typical EC50 values for agonists, in order to identify even low-affinity interactions. To better identify highly active compounds, assays were also carried out at 10 nM concentrations.

Overall, the synthetic cannabinoid library was highly active with the CB1R biosensor (Figure 3a), where the majority of compounds showed over the half-maximal fold-change value at 1 µM. To validate the results, known and predicted agonists and antagonists of the receptors were examined individually. The known antagonist URB447 [31], predicted antagonist WIN54,461 [32], non-binder HU-211 [33], and cannabidiol degradation product HU-311 [34] all showed less than 25% relative fluorescence of the ACEA positive control at CB1R at 10 nM (Figure A3), indicating that the yeast biosensors could readily identify both agonists and antagonists. In contrast, the known agonists AB-FUBINACA, ADB-PINACA, and AB-PINACA showed 79–84% maximal activity at this concentration.

The CB1R biosensor gave consistent characterizations within compound classes. For example, in the PINACA agonist series (AB-PINACA, ADB-PINACA AND 5-fluoro ADB-PINACA isomer 2), all were shown to have EC50 values in the low nanomolar range, except 5-fluoro-2-ADB-PINACA isomer 2 (Figure 3b), which had a unique positioning in the pentyl fluoride on the diazole ring. The drastic loss of activity in this molecule suggests that this structure blocks the molecule from binding the active conformation of the receptor.

Dose response curves for AB-FUBINACA isomers were also obtained, including for many compounds that had previously been uncharacterized. AB-FUBINACA and AB-FUBINACA isomer 1 showed high maximal signaling, while isomers 2 and 5 had a lower response (Figure 3c). Since AB-FUBINACA has been implicated in overdoses from recreational use along with several of its variants [9], knowledge of the activity of structural variants could be useful for the determination of compound scheduling.

### 2.4. High-Throughput Drug Screening of Terpenes with Dual Cannabinoid Receptor Biosensor Strains

It has been hypothesized that THC and many of the terpenes found in cannabis strains work synergistically to create strain-variant effects [35], so we also assessed a library of terpenes against the CB1R biosensor strain (Figure 3d). A number of compounds in the library have been found in cannabis, and others are known to be bioactive [35]. In order to detect even minor activities, compounds were screened at 100 µM, but very few compounds showed activity against CB1R. Amongst the terpenes found to activate CB1R, both (+)-β-Citronellol and β-Eudesmol have previously been isolated from cannabis [36,37], and β-Eudesmol showed signaling at high (10 µM) concentrations (Figure 3e).

### 2.5. Mammalian Cell Assays Validate Hits from Yeast Screening

To validate the findings and further evaluate the performance of our yeast screening assay, we retested a subset of our library in a conventional mammalian cell assay. We acquired human cell lines depleted of native GPCRs and containing the promiscuous Gα15, wherein GPCR activation triggers intracellular calcium release, which is measured by Fluo-8 fluorescence (see Section 4). We selected eight compounds for side-by-side testing between the yeast and human host for CB1R (Figure 4a–d). To maximize the number of compounds tested per plate, we performed a limited four-point dose response on the mammalian cells (Figure 4a,b).

Overall, the comparison showed that the yeast strains were highly sensitive to all cannabinoid agonists validated in mammalian cells. Of the eight compounds, only one (5-fluoro 2-ADB-PINACA isomer 2) failed to induce significant levels of fluorescence at high concentrations (10 μM) in mammalian cells (Figure 4a,b). In the yeast assay, this compound also showed the lowest activity, though it did show partial agonism at high concentrations (10 μM) (Figure 4d). Among the remaining CB1R agonists, the yeast biosensor was consistently orders of magnitude more sensitive than the mammalian strain. EC50 values in the mammalian background were estimated to be between 10^−6^ and 10^−8^ M, while the yeast values ranged from 10^−9^ to 10^−12^ M (Appendix A). Since the human CB1R strain uses the full-length receptor, differences in plasma membrane localizations may contribute to this shift in sensitivity (in addition to the effect of membrane cholesterol and other host-specific effects). Given our limited four-point dose response, it is however difficult to compare rank-order sensitivities between the two assays. Collectively, these findings suggest that the yeast biosensor is likely capable of identifying agonists with high sensitivity; however, differences in rank-order potency or other discrepancies may be expected in mammalian cells due to fundamental physiological differences between hosts. Therefore, we suggest this strain is a tool best used for identifying leads for downstream characterization in human cells. 

Indeed, the mammalian experiments validated several of our hits from the yeast compound screening efforts: AB-BICA (yeast EC50: 4.87 × 10^−6^ M), AB-FUBINACA isomer 1 (EC50: 6.13 × 10^−8^ M), and 5-fluoro PB-22 5-hydroxyisoquinoline isomer (EC50: 2.59 × 10^−7^ M) have not been previously evaluated at CB1Rs to the best of our knowledge (Appendix A).

## 3. Discussion

The cannabinoid receptor CB1R is of clinical interest for its roles in pain relief, appetite stimulation or suppression, and anti-epileptic properties. Additionally, the recreational cannabis industry has also seen enormous growth while newly discovered synthetic cannabinoids are continuously emerging. The speed at which many of these drugs are created makes it difficult for regulatory bodies and health authorities to keep pace, so these substances are often under-researched and pose a risk to user health.

We also sought to further optimize and validate the CB1R biosensor strain by humanizing the host sterols. Previously, it was shown that cholesterol was indeed a negative regulator at CB1 receptors [28], though it was not clear if this effect would be mimicked by ergosterol, which only differs by two double bonds and a methyl group. We observed cholesterol and its intermediates appear to negatively regulate signaling, which indicates that CB1R is relatively specific for cholesterol. The observation that cholesterol negatively regulates CB1R further validates the biosensor as a reliable proxy for native function. 

By engineering CB1R, we were able to generate a biosensor that could be used for the rapid screening of a wide variety of cannabinoids and other compounds, such as terpenes, including dozens of previously uncharacterized compounds. We used mammalian cells to validate a subset of agonists identified in yeast screening experiments, which confirmed the identification of known and new compounds. However, our validation experiments were limited to eight compounds, so more testing is needed to determine the broader applicability of the yeast strain as a surrogate for mammalian cells. Experimental results may be affected by differences in protein folding, post translational modification, pH, growth temperature, regulatory proteins, and plasma membrane composition. In this work, we explored the effect of membrane sterol composition on CB1R functioning in yeast and found that ergosterol did not significantly attenuate signaling compared to cholesterol, but other host-specific effects may exist.

In addition to advancing pharmaceutical drug development, the emerging cannabis industry exposes consumers to a large number of uncharacterized compounds. The sheer speed at which many psychoactive mixtures can be created makes it difficult for regulatory bodies and health authorities to keep pace, highlighting the need for studies that can quickly provide insights into receptor binding and specificity. The availability of facile comparative assays via yeast biosensors builds on earlier work expressing CB2R in yeast [13,14] and may potentially provide a straightforward basis for comparisons suitable for both research and regulatory organizations. Finally, given that the complete biosynthesis of cannabinoids has been achieved in yeast [38], and that yeast biosensor strains expressing GPCRs have begun to show promise in detecting metabolites during bio-manufacturing [17,39], we can readily envision adapting the GPCR-based sensors herein to the conjoined screening and selection of new pathways and receptors that are specific for any of a variety of cannabinoids or other compounds.

## 4. Materials and Methods

### 4.1. Molecular Biology

All cloning was performed using Golden Gate assembly following the MoClo Yeast toolkit [17] with some adaptations (see Parts list). Assemblies were performed as follows: 20 fmol of part plasmids, 10,000 units of Type IIs restriction enzymes (T7 DNA ligases, Esp31/BsaI-v2, NEB, Ipswitch, Burlington, MA, USA), and 1 μL of T4 DNA ligases (NEB, Ipswitch, Burlington, MA, USA) in a 10 μL reaction. Thermal cycling was performed as follows: 1 min at 37 °C and 2 min at 16 °C for 25 cycles, then 37 °C for 30 min, and 80 °C for 10 min. An amount of 5 μL of each reaction was transformed into 100 μL DH10B and transformed according to the Mix and Go Transformation kit (Zymo Research, Irvine, CA, USA).

### 4.2. Yeast Transformations

The yeast background strain was BY4741 (See yeast strain list). Yeast transformations were performed according to the EZ transformation II kit (Zymo research, Irvine, CA, USA). An amount of 100 μL of cell prep was transformed with 1 μg or 5 μL of plasmid.

### 4.3. Yeast Functional Assays

Yeast colonies were picked and grown overnight to saturation in pH 5.8 SD-His media in a 2.2 mL deep well plate (Axygen, Corning, NY, USA), grown in a plate-shaking incubator (30 °C, 1000 rpm, 3 mm orbital). In the morning, cultures were diluted to 1:10 or 1:25 in SD-His media buffered to pH 7.1 with 100 mM MOPSO (unless otherwise specified). Ligands were added to the culture and incubated with shaking for 8 h. Cells were then washed three times with ice-cold Tris buffer (pH 7) and diluted to 1:20 for cytometry. All cytometry was performed on a Sony SA3800 spectral analyzer. Each sample was tested for 10,000 events and read at a rate of 1000 events per second. All assays as reported here were filtered only for singlet populations. For compound library screening, the Cayman Chemical Synthetic Cannabinoid library (cat. no. 9002891) and Terpene library (cat. no. 9003370) were used (Cayman Chemical Scientific, Ann Arbor, MI, USA).

### 4.4. Mammalian Functional Assays

Mammalian cell assays were performed in Eurofins Ready-to-Assay™ CB1 Cannabinoid Receptor Frozen Cells. Cells were division-arrested, which limited the number of samples that could be measured, so a limited 4-point dose response was performed. Cells were seeded in 96-well plates and grown for 24 h. Cells were washed with Hank’s Balanced Salt Solution (HBSS) with 20 mM HEPES and 2.5 mM Probenecid at pH 7.4. An amount of 1 mg of Fluo-8 AM calcium dye (AAT Bioquest cat. no. 21080, Pleasanton, CA, USA) was resuspended in 200 μL DMSO and then diluted to 1:1000 into the HBSS 20 mM HEPES and 2.5 mM Probenecid buffer and added to the cells. Cells with dye were incubated at 37 °C for 1 h before fluorescent measurement on a Flexstation 3-plate reader (Molecular Devices, San Jose, CA, USA). Baseline fluorescence was measured for 10 s before 50 μL ligands were injected into 100 μL cells. Samples were measured for 180 s and recorded values were measured as ΔF/F_0_ (Fmax − Fmin/Fmin).

## 5. Patents

E.C.G., J.D.G., A.D.E., C.J.M., E.M.M., and J.L. have filed a provisional patent application on the yeast-based CB1R biosensor, USPTO Application 63/285,337.

## Figures and Tables

**Figure 1 ijms-25-06060-f001:**
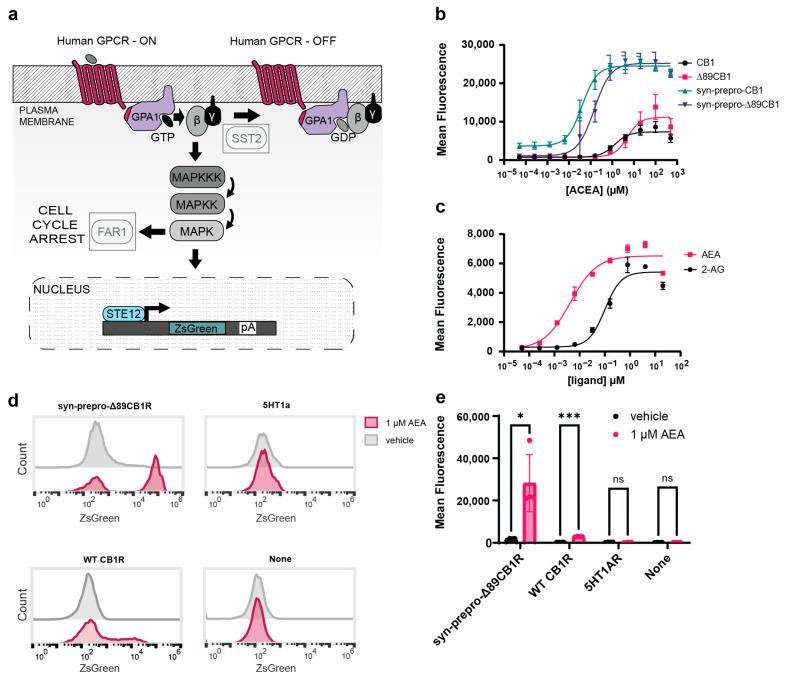
Development of the CB1R biosensor. (**a**) The yeast pheromone response pathway was genetically modified for human receptors. Grayed out boxes represent knocked out genes. The yeast pheromone GPCR Ste2 was replaced with a human GPCR (illustrated as a red 7-transmembrane protein). The Fig1 pheromone response gene was replaced with a ZsGreen reporter via knock-in. The GpaI G-alpha protein is illustrated in two colors to represent its chimeric nature, where the yeast portion (purple) interacts with the factors of the MAP kinase cascade and the humanized C-terminus (red) interacts with the human GPCR directly. (**b**) ACEA dose response with CB1R. CB1, EC50 = 1.30 µM, 95% CI 0.80 to 2.12 µM. Δ89-CB1R, EC50 5.05 µM, 95% CI 2.64 to 9.29 µM. Syn-pre-pro CB1R EC50 = 0.04 µM, 95% CI 0.03 to 0.05 µM. Syn-pre-pro Δ89-CB1R, EC50 = 0.15 µM, 95% CI 0.10 to 0.21 µM. (**c**) Dose response of the endocannabinoids 2-arachidonoyl glycerol (2-AG) and N-arachidonoylethanolamine (anandamide, AEA) with CB1R. AEA, EC50: 4 nM, 95% CI 2.62 × 10^−9^ to 6.92 × 10^−9^ M 2-AG,EC50: 100 nM, 95% CI 6.60 × 10^−8^ to 1.48 × 10^−7^ M. For all experiments, *n* = 3. Nonlinear regression was fit as a 4-parameter logistic equation. (**d**) Cytometry of yeast treated with 1 uM AEA or DMSO vehicle (1% *v*/*v*). The yeast was treated with ligands or vehicles for 8 h, and ZsGreen fluorescence was measured on gated singlets. Syn-pre-pro-Δ89CB1R (*n* = 4), WT CB1R (*n* = 4), 5HT1AR (*n* = 3), and the base GPCR strain with no receptor (*n* = 4) were tested. (**e**) Mean fluorescence values from (**d**) for the AEA and vehicle group were plotted. Unpaired *t*-tests were performed between the negative control and test groups. *p*-values are reported where ns corresponds to *p* > 0.05, * is *p* ≤ 0.05, and *** is *p* ≤ 0.001.

**Figure 2 ijms-25-06060-f002:**
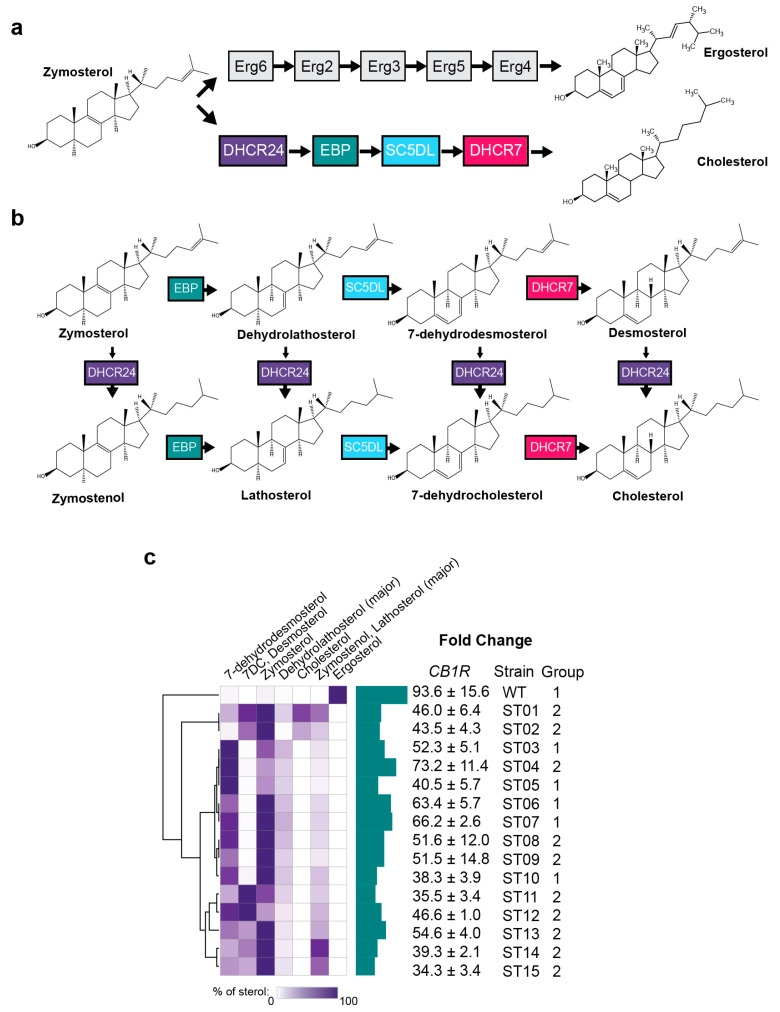
Optimization of sterol environments for CB1R. (**a**) Comparison of ergosterol and cholesterol production pathways in yeast and humans, respectively, after the common intermediate zymosterol. (**b**) Cholesterol biosynthetic intermediates and the enzymes that act on them after zymosterol. (**c**) Fold change in signaling results for CB1R in sterol-modified strains. Strains were engineered to produce cholesterol and\or cholesterol biosynthetic intermediates, and the sterol content was measured using GC-MS in a previously published work reproduced here with permission. Dose responses were performed for both receptors in each strain, and fold changes of the maximum signal-over-signal with 0 µM agonists were calculated. The agonist ACEA was used for CB1R. For all experiments, *n* = 3. Nonlinear regression was fit as a four-parameter logistic equation.

**Figure 3 ijms-25-06060-f003:**
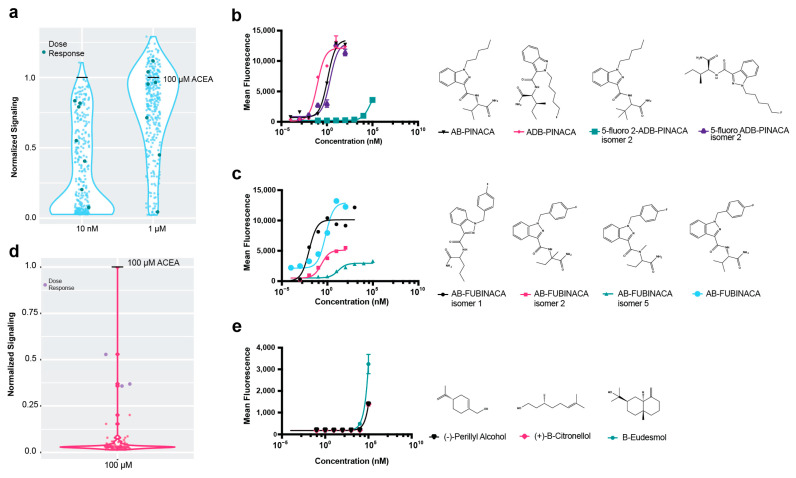
Screen of cannabinoid receptors with compound libraries. (**a**) Screen of CB1R with synthetic cannabinoid compounds at 1 µM and 10 nM concentrations. Signaling was normalized to signaling with 100 µM ACEA, set at “1”. “Dose response” compounds were those chosen for further analysis by dose response. (**b**) Dose responses with FUBINACA compounds are shown. The CB1R yeast strains with the ZsGreen reporter were induced with ligands for 8 h and measured by cytometry. AB-FUBINACA, EC50 = 0.66 nM, 95% CI 0.35 to 1.14 nM; AB-FUBINACA isomer 1, EC50 = 0.01 nM, 95% CI 0.01 to 0.02 nM; AB-FUBINACA isomer 2, EC50 = 0.23 nM, 95% CI 0.1356 to 0.42 nM; AB-FUBINACA isomer 5, EC50 = 15.86 nM, 95% CI 7.63 to 38.36 nM. (**c**) Dose responses with PINACA compounds. AB-PINACA, EC50 = 1.21 nM, 95% CI 0.96 to 1.52 nM; ADB-PINACA, EC50 = 0.09 nM, 95% CI 0.05 to 0.14 nM; 5-fluoro ADB-PINACA isomer 2, EC50 = 2.25 nM, 95% CI 1.26 to 3.97 nM. (**d**) Signaling at CB1R with terpene compounds at 100 µM. Signaling was normalized to signaling with 100 µM ACEA, set at “1”. “Dose response” compounds were those chosen for further analysis by dose response. (**e**) Dose responses at CB1R with compounds that demonstrated signaling in the terpene screen over 0.25 greater than that of the 100 µM ACEA control. The low-affinity terpenes did not show fluorescence except at high levels of the agonist. For all dose response experiments, *n* = 3. Nonlinear regression was fit as a four-parameter logistic equation. For the screens, *n* = 1.

**Figure 4 ijms-25-06060-f004:**
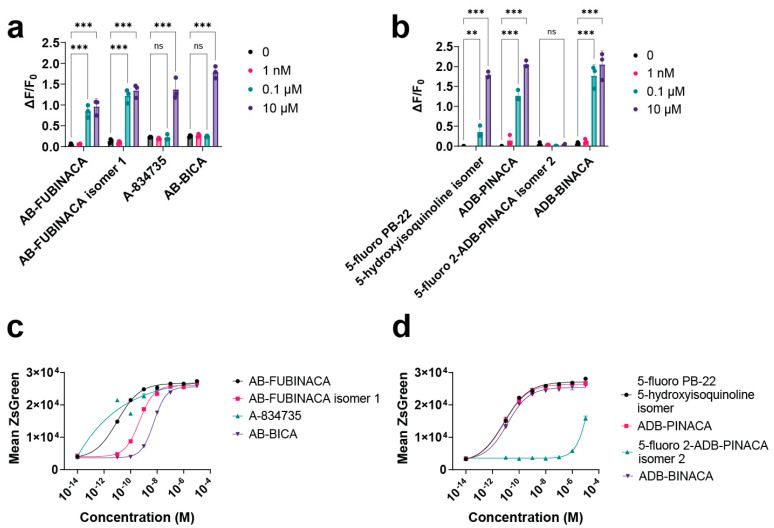
Mammalian cell experiments (**a**,**b**) were performed in a commercial cell line with a promiscuous Gɑ15 that released intracellular calcium upon GPCR activation. Cells were injected with ligands, and the fluorescence of a calcium dye was measured using a plate reader. Change in fluorescence was measured as ΔF/F0 (Fmax − Fmin/Fmin). Yeast dose response experiments (**c**,**d**) were performed as previously described using cytometry to measure the ZsGreen transcriptional reporter. For all dose response experiments, *n* = 3. A two-way ANOVA was performed between indicated test groups. *p*-values are reported where n corresponds to *p* > 0.05, ** is *p* ≤ 0.01, and *** is *p* ≤ 0.001.

## Data Availability

Data not included in this manuscript are available upon request.

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
