# Peer review of "A Humanized CB1R Yeast Biosensor Enables Facile Screening of Cannabinoid Compounds"

_ijms, 2024, doi:10.3390/ijms25116060_

Round 1

Reviewer 1 Report

Comments and Suggestions for Authors

Natural and synthetic cannabinoids bind to G protein coupled receptors (GPCR) in human cells, and they can be used for recreational and medicinal purposes. Given the role of cannabinoids in the treatment of chronic pain, epilepsy, and psychiatric disorders, ideal therapeutic candidates should activate cannabinoid receptor type 2 (CBR2), but not the type 1 receptor (CBR1). Yeast strains expressing these human receptors could be used to analyze the activity of cannabinoid compounds, and CBR2 yeast strains have been constructed for this purpose. This study generated yeast strains expressing human CB1R and tested the affinity of various cannabinoids to CB1R using this humanized yeast biosensor. Moreover, human cells were used to validate the results from yeast biosensor, indicating that the humanized CB1R yeast strains can be used for the screen of cannabinoid compounds. However, some issues need to be addressed prior to publication.

- More explanation of the construction of yeast CB1R biosensor (Line 72-81) is necessary. For example, it seems that yeast Gpa1 is a yeast membrane protein that is used to anchor CB1R to yeast membrane. It is unclear why the last 5 C-terminal residues of Gpa1 are replaced by CB1R. Similarly, the introduction of human Gai3 and ZsGreen is not sufficient for readers to understand the rationales for the experimental design.

-To test the constructed yeast CB1R biosensor, a yeast strain without CB1R expression needs to be included as a control.

-Figure 1A, the protein with transmembrane domain (red) needs to be labeled (Gpa1?).

-As yeast CB2R biosensors have been constructed in previous studies. It will be interesting to know if these strains show distinct responses to cannabinoid compounds compared to yeast CB1R strains.

-The paragraph (line 236-241) is difficult to follow. It should be reworded.

Comments on the Quality of English Language

Some parts of the manuscript as indicated in the comments above need to be improved for clarity.

Reviewer 2 Report

Comments and Suggestions for Authors

The manuscript by Mulvihill et al. presents the development of a biosensor yeast strain that expresses a humanized CB1R and can be used for rapid screening of synthetic cannabinoids activity. While such CB2R yeast strains have been developed, the new contribution is the functional CB1R strain. The study is significant, but requires edits, mostly clarifying, to be published.

Major:

- Methods should be better described. While small technical details are provided, a more structured explanation of what is precisely being measured and the methodologies employed would enhance understanding. For example, what is bulk fluorescence and what does it reflect, etc. Not only this is essential for understanding and possibly replicating the approach, but this would help in understanding how these measurements correlate to the biological activities of cannabinoids.

- The difference in response between yeast and mammalian cells is observed by the authors. There are inherent limitations in using yeast as a model system that could affect the pharmacological profiling of cannabinoids. A primary concern is that the validation of the yeast biosensor's effectiveness involved a side-by-side screening of only eight, with one of these compounds failing. This limited dataset significantly undermines the study's robustness and does not provide enough statistics to confidently support the authors' claims. More extensive comparative analysis is necessary to establish the reliability and relevance of the yeast biosensor findings in a human biological context.

- Discuss limitations of the study.

- Fig. 3e - does not look like a dose-response - aggregation? ligand-depletion? provide experimental details

Minor:

- some grammar/formatting here and there and clarification is needed. Examples (not a comprehensive list):

l131 metabolites24 - correct ref

l140-141 - difficult to read and understand how the authors draw this conclusion

l161-162 - explain the inclusion of DMSO (conc, of ctrl as well as what was the conc introduced by the compound)

l237-238 and throughout check formating for powers, e.g., 10-7 vs 10-7
